# Evolutionary Study of the Crassphage Virus at Gene Level

**DOI:** 10.3390/v12091035

**Published:** 2020-09-17

**Authors:** Alessandro Rossi, Laura Treu, Stefano Toppo, Henrike Zschach, Stefano Campanaro, Bas E. Dutilh

**Affiliations:** 1Department of Biology, University of Padova, 35131 Padova, Italy; alessandro.rossi.23@phd.unipd.it (A.R.); stefano.campanaro@unipd.it (S.C.); 2Department of Molecular Medicine, University of Padova, 35131 Padova, Italy; stefano.toppo@unipd.it; 3Department of Biology, University of Copenhagen, 1017 Copenhagen, Denmark; henrike.zschach@bio.ku.dk; 4CRIBI Biotechnology Center, University of Padua, 35131 Padova, Italy; 5Institute of Biodynamics and Biocomplexity, University of Utrecht, 3508 Utrecht, The Netherlands; bedutilh@gmail.com

**Keywords:** metaviromics, gene evolution, crAssphage, mirrortree, human gut

## Abstract

crAss-like viruses are a putative family of bacteriophages recently discovered. The eponym of the clade, crAssphage, is an enteric bacteriophage estimated to be present in at least half of the human population and it constitutes up to 90% of the sequences in some human fecal viral metagenomic datasets. We focused on the evolutionary dynamics of the genes encoded on the crAssphage genome. By investigating the conservation of the genes, a consistent variation in the evolutionary rates across the different functional groups was found. Gene duplications in crAss-like genomes were detected. By exploring the differences among the functional categories of the genes, we confirmed that the genes encoding capsid proteins were the most ubiquitous, despite their overall low sequence conservation. It was possible to identify a core of proteins whose evolutionary trees strongly correlate with each other, suggesting their genetic interaction. This group includes the capsid proteins, which are thus established as extremely suitable for rebuilding the phylogenetic tree of this viral clade. A negative correlation between the ubiquity and the conservation of viral protein sequences was shown. Together, this study provides an in-depth picture of the evolution of different genes in crAss-like viruses.

## 1. Introduction

Metagenomics, i.e., the discipline focused on studying genomic sequences from environmental samples, is a relatively new field. The first occurrence of the word itself in the literature dates to 1998 [1]; over the last two decades metagenomics, fueled by the rise in computing power and the development of high-throughput sequencing techniques, has become a major force behind the development of microbiology and environmental biology [2,3]. This revolution is happening for viruses too, whereas the discovery and isolation of new viruses was traditionally linked to the infection of cultivated bacteria, nowadays novel viral sequences are routinely discovered via analysis of environmental samples [4,5]. Thanks to these approaches, the understanding of phage biodiversity has been widely enlarged, with new clades of human- and animal-associated jumbophages and megaphages and novel viruses (without known isolates) being recently described [6,7]. The bulk of the newly discovered viral species is such that the International Committee for the Taxonomy of Viruses (ICTV) has proposed to allow for the classification of new species on the basis of sequence data alone [8]. In summary, metagenomics has given the possibility to look at the microbiological world from a perspective which transcends the bias associated with the methods previously used. These findings led also to a more-comprehensive analysis of the microbial communities and to a more detailed evaluation of the diversity, prevalence, and ecosystem distribution of phages.

One of the most exciting discoveries from metaviromics is the existence of the crAssphage bacteriophage, found in about half of the human population and being the reference for up to 90% of the viral reads in human gut metagenomes [9,10]. It has subsequently been found, in association with human feces, in every part of the world [11]. This viral species, despite its ubiquity, has been discovered only recently, this is due to the divergence of the crAssphage genomic sequence from other viral genomes and the difficulty of growing its host, a species of the Bacteroides genus, in vitro. A method based on single assembly from more than one metagenome, named cross-Assembly (hence the phage’s name) was used for this purpose [12]. An additional study proved that crAssphage is a member of an entirely new viral clade [13]. The putative crAss-like family has been divided into four subfamilies and ten candidate genera [14]; it includes viruses from diverse ecological niches such as termite gut and marine sediments, as well as another known member of the human gut, the immunodeficiency-associated stool virus (IAS). Given the importance of the microbial communities with regards to human health, it makes sense to investigate every aspect of the most abundant virus of one of the most abundant bacteria residing in the human gut. Furthermore, crAssphage’s association with humans is not recent: members of the same viral family, even from the same putative subfamily, are present in non-human primate guts [11]. This suggests that crAss-like viruses are long-time companions of the human lineage. Among the many reasons to study phages, the fact that they can kill specific microbes and can transfer antibiotic resistance or pathogenicity and, consequently, alter host metabolism is one of the most relevant [15,16]. In particular, the impact of variations in the gut microbiome, and potentially of the associated virome as well, on human health is becoming evident [17,18,19]. Although microbial correlations with human pathologies have been observed, no significant association of crAssphage genomic features with health or disease was found [11]. However, as major components of the human gut, crAss-like viruses deserve more attention, and the dynamics among dominant and rare populations recently described shall be further investigated [11]. In this respect, understanding the overall crAssphage genome evolution and the evolutionary relationships between its individual genes may provide new information.

The crAssphage genome is composed of a dsDNA 97 kbp long circular sequence, divided in two sections in which genes have different functions: replication-related genes are found in the forward strand, while all the other genes are encoded in the reverse strand [9,13]. Similarly, transcription and capsid-related proteins are found in single-purpose blocks of the genome. This organization is common among viruses, and particularly bacteriophages; in fact, this tendency is used in order to identify prophages amidst bacterial scaffolds [20]. Not all genes fit in this model, though. Despite all the efforts, the gene functions in a vast part of the genome remain unknown. Thus, evaluating the coevolution between interacting protein families could provide additional insights on their putative biological function. Specifically, host–parasite interactions are known to be deeply influenced by coevolution, as well as it is known to influence conspecific populations that may diverge or co-adapt under different circumstances [21,22]. Evaluation of the similarity between proteins evolutionary histories has been successfully achieved by correlating mutations using multiple sequence alignments [23]. In this respect, while the previous studies mainly focused on the crAss-like genome as a whole, in the current work we performed a gene-centric analysis of crAssphage evolution. In particular, we examined the conservation degree of protein sequences and the degree of coevolution among them, as well as the relation to their functional roles. The overall aim of our study is to measure the frequency and ubiquity of crAssphage proteins divided by function in order to define a global vector of conservation.

## 2. Materials and Methods

### 2.1. Sequences Selection, Retrieval, and Data Preparation

A database of 805 crAss-like assembled sequences was created (Appendix A
Appendix A), retrieved from previous works [11,13]. The reference crAssphage genome was included in the database. Redundant copies of the same sequences were removed with custom software developed using the Python programming language version 3.6.3 [24] and the Biopython package [25]. PRODIGAL v.2.6.3 [26] was used to predict the genes for all the crAss-like contigs in the database, using the meta-procedure appropriate for viruses and small plasmids. Afterwards, the proteins predicted from the reference crAssphage genome were used to recover the homologous proteins from all the crAss-like contigs (Appendix A). For this step, each protein was used as a PSI-BLAST [27] query against the whole crAss-like contig database, with maximum e-value of 0.001 and minimum query coverage of 80% to minimize potential domain walking artifacts. In order to validate the robustness of the PSI-BLAST homology detection, the search was repeated with the query coverage percentages of 50% and 95%. The PSI-BLAST searches were run until convergence. The annotation and function of crAssphage reference genes were retrieved from a previous study [13]. The ORFs of the previous and current studies were compared by BLASTp [28] search considering 90% of query coverage as lower threshold.

The protein functions were loosely grouped into six functional modules, following the spatial division along the reference crAssphage genome, as previously proposed [13]. The functional groups are “Uncharacterized”, in which all the proteins with unknown function are grouped, “Replication”, “Transcription”, “Tail and structural”, “Capsid”, and “Other”, which comprises four proteins whose function has been identified but do not fall into any of the previous categories. The amino-acid sequences were grouped into clusters for each reference protein they matched in the PSI-BLAST search, resulting in 92 clusters, one for each reference crAssphage ORF as predicted by PRODIGAL. Only contigs which included two or more ORFs were included in the database, and this filtering lowered the number used in this study from 805 to 370. Protein sequences that were duplicated or had introns in the reference genome were treated separately. Three different clusters were created for the two RepL genes: one cluster comprises all the genes that hit only the Reference_crAssphage.1_45 reference ORF, another comprises all the genes that hit the Reference_crAssphage.1_91 ORF and the last one includes both groups. The clusters were aligned using MAFFT v7.271 [29] with standard parameters and the L-INS-I algorithm.

### 2.2. Phylogenetic Tree Reconstruction and Comparison

For every homologous group of proteins a ML tree was built with IQ-TREE v1.6.1 [30] with standard settings and 1000 replicates ultrafast bootstrap [31]. For this analysis, the dUTPase homologous group was manually split into two halves, using AliView v1.19 [32]. Despite the difference in length among the sequences in the RepL homologous group, the alignment did not need manual trimming, as it does not present badly aligned regions which would impair the phylogenetic reconstruction. Furthermore, the phylogenetic tree reconstruction software treats gaps as not containing information, and as such the presence of a well-aligned subset of longer sequences does not have an impact on the overall quality of the reconstruction. The ML matrices generated by IQ-TREE were used for the comparison of phylogenetic trees. Matrices were compared pairwise with the Mirrortree method [33]. In each comparison, rows and columns were trimmed in order to keep only the nodes coming from the same contigs; the matrices were then converted into vectors and compared using Pearson correlation. Pairs of homologous groups which shared less than 5 sequences coming from the same genome were discarded, in order to avoid biases due to low sample size. These operations were performed via custom scripts in Python, using the Scipy package [34].

### 2.3. Quantifying Sequence Conservation

The conservation of the amino acid sequences was measured as the average Shannon information content across the entire alignment of sequences, as defined by Shannon [35], via custom Python scripts. As a comparison, all Prokaryotic Viral Orthologous Groups (pVOGs), which include proteins coming from viruses belonging to the Podoviridae family, were retrieved in May 2020 [36]. The annotation was retrieved and used to classify the proteins according to their functional categories, using a custom Perl script and based on crAssphage categorization [13]. They were aligned as described above and the Shannon information content was calculated using the same scripts. Linear correlation (R and *p*-values) between Shannon information content and log10 value of the number of genes in the group was calculated independently per each functional class and plotted using ggpubr R package.

### 2.4. Function Prediction and Distant Similarity Searches

To further characterize the proteins assigned to the “Uncharacterized” group, recent and best performing algorithms for protein function prediction were applied; these include the sequence-based Argot2.5 server [37] and 3D structure-based I-TASSER [38]. Both methods have been chosen according to their recent best performance in community-wide challenges of prediction methods CAFA3 [39] and CASP13 [40].

### 2.5. Data Visualization and Availability

All the data visualization was performed using the Matplotlib package for Python [41], embedded in various in-house developed scripts. Phylogenetic trees were displayed via the iTOL v4.42 website [42]. All the in-house developed scripts, as well as the complete dataset including all proteins, annotations, and homologous groups are publicly available on a GitHub repository https://github.com/Ale-Rossi/crAssphage-gene-evolution.

## 3. Results

### 3.1. Protein Identification and Clustering

Whereas the length of the crAssphage reference genome is 97 kbp, only 5% of the contigs (41) lie in the 80–100 kbp range; thus, most of them do not represent complete genome sequences. Eighty-one percent of the contigs used in this study (302) are shorter than 40 kbp and the average length is 26 kbp (Appendix A). While the high number of shorter contigs is probably due to incompletely assembled genomes, we were still able to use these sequences to quantify the co-evolutionary signal between protein families that are encoded on the same unit. Moreover, some long contigs share relatively few ORFs with the crAssphage reference genome, representing distantly related crAss-like bacteriophages [13,14].

A total of 92 ORFs were identified in the reference crAssphage genome and, using similarity search against ORFs identified in all other crAss-like genome sequences, 92 protein clusters were produced. Most proteins were assigned to a single cluster, with two notable exceptions as described in the Methods section and further analyzed below. Although the Prodigal software is not optimized for viral genomes, it identified the same genes as previous studies that used Glimmer and MetaGeneMark [9,13]. More specifically, the first study identified 80 protein-encoding genes; the latter identified 90 protein-encoding genes. We further classified the proteins into functional groups: the replication and structural modules are the largest, featuring 24 and 17 genes, respectively; the capsid, replication and “other” groups are very small, with only 3, 2, and 4 genes respectively. No annotation was available for 42 genes representing the “Uncharacterized” group. The predictions of this group and a tentative consensus between Argot2.5 and I-TASSER showed an improvement in annotation and partial agreement between the two tools for 4 out of 42 proteins (Appendix A).

Each protein cluster comprises from 8 to 174 sequences, not equally distributed across the functional categories. In fact, some clusters have a less variable number of sequences, partly due to the linkage of genes closely located on the genome. Nonetheless, all the functional groups have a median of more than 50 sequences per cluster. Additionally, most of the proteins with a low cluster size are uncharacterized (Figure 1). The capsid proteins stand out as the most ubiquitous ones, with the major capsid protein (MCP) being the most widespread gene, and present in 174 contigs.

Several proteins presented exceptions to the standard clustering applied in this work. The gene encoding RepL is found in two copies in the crAssphage reference genome. The paralogs arose through an ancient gene duplication, as supported by the RepL phylogenetic tree, which shows a distinct separation between the two ORFs (Appendix A). In addition, a total of three gene duplications have been newly detected in the collection of crAss-like phage contigs, including duplications of Reference_crAssphage.1_44, Reference_crAssphage.1_73, and Reference_crAssphage.1_74. In particular, Reference_crAssphage.1_73 encodes for a tail sheath protein that, following the reference genome structure, is included in the short module composed of tail and structural proteins. Six contigs contain duplications of the Reference_crAssphage.1_74 sequence (i.e., Gut.14, Gut.03, Gut.07, Gut.05, Activated_sludge.3, and Gut.06). In the phylogenetic tree of the protein, these sequences are very closely related (Appendix A). No annotation is available for this ORF, but the genomic context and the presence in close proximity of genes encoding capsid proteins are suggestive of a structural role. Additional evidence comes from the iVIREONS structural protein score (0.62) [9], and putative structural homology to a kinase (max TM-score: 0.62, max C-score: −3.71) according to I-TASSER [38] (Appendix A). RepL proteins, which were first found in *Staphylococcus aureus* plasmids, are known to increase the number of copies of the plasmids they are found in [43]; this prompts us to speculate that the duplication could have given the virus an evolutionary advantage relative to a heightened reproductive ability. In-depth studies are needed to confirm this. There is not enough data to allow for speculation regarding the other instances of gene duplication, as most events are found in single genomes, and it is then not possible to predict how the duplications play a role in the phage life cycle.

Additionally, another protein cluster contains two ORFs encoded on the crAssphage reference genome recognized as two distinct regions of a dUTPase gene that are split by the insertion of an intron-encoded endonuclease belonging to the HNH protein family, Reference_crAssphage.1_35. The intron insertion is relatively recent and was found in nine contigs, all of which were assembled from shotgun reads collected from gut samples of the twin sisters of a single family [44]. This can also be seen in the phylogenetic tree where the sequences having the insertion form a single clade (Appendix A).

### 3.2. Protein Conservation

In order to improve interpretation of the findings in the crAss-like family, all the prokaryotic viral orthologous groups (pVOGs) were analyzed and trends in the distribution of the conservation degree were identified. In pVOGs, there is a negative correlation between the number of sequences in the alignment and the Shannon information content (Figure 2). The relation between the two variables can be described with a logarithmic regression with the correlation coefficient ranging from −0.6 to −0.81 according to the functional category (*p* < 0.05 for all categories). The pVOGs as a whole show a correlation coefficient of −0.67 (*p* < 2.2 × 10^−16^). A similar trend, albeit not as pronounced, can be seen in crAss-like protein homologous groups too. Both the structural, the uncharacterized and the replication proteins have a negative correlation coefficient, but the *p*-value calculated for the latter category was not significant (R = −0.41, *p* = 0.0073; R = −0.65, *p* = 0.0049; R =−0.29, *p* = 0.19). The remaining functional categories, having a very small size, were ignored. crAss-like proteins, as a whole, show this correlation as well (R = −0.39, *p* = 1 × 10^−4^). The most likely explanation for this negative correlation is a sampling effect where many, widespread sequences may contain more diversity than few sequences with a narrow occurrence distribution. The stronger correlation coefficients of pVOGs relative to the homologous groups of proteins in crAss-like viruses that were built in this study is likely due to other factors. Firstly, pVOGs are built from proteins found across all the prokaryotic viruses and include several homologous groups with a wide distribution that strongly contribute to the low correlation coefficient (note the difference in scale of the X-axes in Figure 2). Conversely, the homologous groups of proteins built in this study only include sequences from the relatively closely related group of crAss-like viruses, which have been proposed to represent a single viral family. Thus, no homologous groups are observed with more than a few hundred sequences and the correlation coefficient is less extreme than for the pVOGs. Secondly, the high threshold of 80% query coverage used in the BLASTp has raised the degree of conservation between the detected homologous proteins that formed the crAss-like clusters. For this same reason, the average Shannon information content of crAss-like proteins is rather high; both the mean and the median lie between 3.0 and 3.5 bits, compared to a maximum of about 4.32 bits. The variation in information content in proteins is consistent across almost all the different protein functions (Appendix A). In all the six groups of proteins we found both highly conserved and divergent sequences, the exception being the capsid proteins. The capsid proteins are unusually low in Shannon information content, as their average value is just 1.95 bits. As with other widely spread genes, it is possible that their ubiquity has led to a great divergence, and that there is less negative evolutionary pressure on them.

### 3.3. Tree Comparison: Consistency of Evolutionary Signal

In order to measure to what extent the encoded proteins followed a similar evolutionary history, we ran a Mirrortree algorithm among pairs of homologous groups of proteins, which consists of the correlation between the distance matrices derived from each pair of phylogenetic trees. The distribution of the 4186 pairwise Pearson coefficients follows a two-peaks distribution: the higher peak ranges from around 0.7 to 1 and consists of 357 pairwise correlations, while the lower peak is from −0.1 to 0.4, including 1409 pairs (Figure 3). It can be seen that there are regionally defined groups of highly correlating genes distributed in the genome (Figure 3). The first of these regions (gene position 12–28) encompasses part of the replication module, including the DNA polymerase, a helicase, a primase, and a DNA ligase, i.e., the core of the DNA replication machinery, as well as enzymes involved in the protection of DNA, such as an uracil-DNA glycosylase and a thymidylate synthase. The second large region (gene position 72–88) including high-correlation genes ranges from the end of the structural to the “other” functional module, encompassing all the capsid encoding genes. Other, narrower regions include transcription and structural proteins (gene positions 31, 33, 47–48, 51–54, and 58–59). These results suggest the presence of a subset of genes evolving coherently with each other while others undergo a different evolutionary history, a possible cause of which is genomic mosaicism, which is common in bacteriophages. Overall, the distributions of average correlation coefficients across the functional groups is relatively even (Appendix A); all the functional categories feature highly correlating genes, and the three most abundant groups (uncharacterized, replication, and structural) all feature low-correlating genes (Appendix A). The capsid proteins, though, stand out as the group with the highest correlations, since none of their average mirrortree coefficients are lower than 0.6 (*p* = 0.0039, one-tailed Mann–Whitney U test).

## 4. Discussion

Genomes exhibiting mosaicism are composed of parts with different evolutionary origins and history. This phenomenon is particularly evident in viral genomes, due to evolutionary processes such as horizontal gene transfer and recombination both with the host and other viral species. In fact, their evolutionary model has been described as the accretion and exchange of various genetic modules [45]. We propose that mosaicism is a prominent feature of crAss-like viruses as well. The heatmap representing the pairwise correlation coefficients (Figure 3) bears witness to this observation: the presence in the genome of regions with a high degree of coevolution, detected with the mirrortree coefficients that were calculated from pairs of phylogenies of homologous groups of proteins, can be interpreted as a result of these evolutionary mechanisms, with the low-correlating genes being accessories to a core genome.

Furthermore, the protein prediction and clustering steps reveal the existence of crAss-like viruses sharing only a small fraction of genes with the reference genome, these genes belonging mostly to the capsid and structural functional modules.

Our approach and the cutoffs used for detecting homologs were rather strict, as nearly full-length homology was required (50%, 80%, and 95% of the query length). Based on these cutoffs and the dataset used, the capsid proteins were found to be the most widespread proteins in the crAss-like viruses. They have been used as phylogenetic markers by Yutin and colleagues because of this very reason [13]. In fact, they are among the genes typically used as signature genes for the identification of viral sequences in metagenomic samples and phylogenetic trees reconstruction. Other genes frequently used as such are portal proteins, tail sheath proteins and polymerases, and specific metabolic genes [46]. Their sequence conservation is below average; although surprising, this could reflect the variety of crAss-like phage species they are found in. After all, an inverse correlation between sequence conservation and spread seems to be the norm, as seen in crAss-like genomes and pVOGs alike. Whereas all the categories feature proteins with high average correlation coefficients, the capsid proteins are by far the group with the highest statistic; this observation highlights how crucial these genes are in phage evolution. From both observations it emerges that capsid proteins in crAss-like phages are confirmed to be one of the most important protein families when it comes to reconstructing their evolutionary history [47]. In summary, whereas other proteins could be regarded as either less plastic/adaptable than capsid proteins, or having too much variability, capsid proteins show themselves as among the most malleable of the crAss-like phage proteins. Indeed, these proteins may be able to reflect the evolution of crAssphage and to develop variation without losing functionality.

It would be easy to speculate that the genes which make up the most phylogenetically consistent part of a viral genome are well conserved, but this is not the case. For each protein cluster the mirrortree statistics done in comparison with every other cluster was averaged. In none of the functional groups was it possible to find a correlation between the average mirrortree statistics and the Shannon information content of a homologous group of proteins. Nonetheless, there can be a discrepancy between the species tree and the gene trees: overall the capsid proteins are probably the best way to retrieve the species tree [47]. Nevertheless, they have been confirmed to be the most ubiquitous among all genes, being present in 174, 155, and 143 crAss-like contigs each. The only homologous groups featuring a similar number of sequences, i.e., 160 and 164, belong to uncharacterized proteins whose genes are close to the capsid-encoding genes in the reference genome. Despite being not characterized, they had already been theorized as capsid proteins [13]. This finding invigorates this hypothesis. The capsid proteins are actually an excellent example of how genes can have a very consistent evolution with other genes while keeping a low level of sequence conservation: this could be due to a similarity in tree topology, with branch lengths varying proportionately. Moreover, it is possible that the higher variability and high average correlation coefficient of these proteins are due to the larger number of homologs; the high correlation statistic would emerge from the subtrees being consistent with the trees of other proteins. On the one hand, it would be assumed that genes which lie on the same genome show similar evolutionary patterns; on the other hand, the mirrortree method was developed in order to identify interacting proteins, not necessarily from the same organism. Nevertheless, many of the proteins within an organism do interact reciprocally: structural proteins interlock with each other in order to form the virion’s structure, and proteins that build complexes similarly do. However, different evolutionary pressures acting on different genes (and different regions on the same gene), and phenomena such as recombination, transposition, horizontal gene transfer, and gene duplications often lead to different genes of the same genome having trees with different topologies [48,49,50]. In light of this, such a high degree of coevolution is definitely remarkable in the evolution of crAss-like viruses.

No functional prediction is currently available for almost half of the genes in the reference genome. There is good reason to believe that many of these genes are of great interest, when studying the biology of crAss-like viruses: some uncharacterized genes are ubiquitous in the viral contigs. Another important finding to point out is that, among the proteins which have a high mirrortree coefficient with other proteins, all the functional groups are represented. These protein categories only loosely share a function, so, with some hindsight, it is not surprising to attest how much they diverge. It is worth noticing, though, that even among the top-scoring proteins the uncharacterized category is represented, but including also highly conserved proteins. This is the same for every group, with the exception of the capsid proteins. This means that many genes are highly conserved and strictly co-evolving with other genes, thus potentially interacting with other genes and important in the virus biology, have an unknown function. The uncharacterized functional group in the crAssphage genome is by far the most numerous among the six categories, boasting 42 genes, whereas the second most abundant one only has 24.

The difficulty of annotating genomes is a huge problem when trying to decipher viruses’ biology. In this study, we attempted to annotate these proteins with available top-performing function prediction tools. Unfortunately, all the proteins revealed to be difficult targets and only for a small portion of them it was possible to extract functional predictions from Argot2.5 and I-TASSER. In spite of the scores not being very high, results provided potential function for four proteins, i.e., Reference_crAssphage.1_12, Reference_crAssphage.1_13, Reference_crAssphage.1_50, and Reference_crAssphage.1_62 (Appendix A). This may be, indeed, due to many factors such as the short length of some proteins submitted for the prediction that could be artifacts of the gene prediction step. One brute-force approach to deal with the lack of annotation could just be ignoring all the uncharacterized proteins and focus our attention only on the already annotated ones. In fact, the 89 ORFs previously reported [13] are only slightly different from the 92 genes identified in the present study. The additional sequences are very short and are not matched by other viral genes in the BLAST search. In fact, many of the homologous groups of proteins with the lowest sequence count belong to the “Uncharacterized” functional group (Figure 1).

One of our findings is that crAss-like family phages follow the trend of prokaryotic viruses in which more widespread genes appear to be less conserved, with the capsid proteins being the utmost representatives of this situation. crAssphage’s genes are clearly consistent in their evolution: there is a bulk of genes showing a high coefficient of coevolution. The capsid proteins are confirmed to be a good choice when building a viral phylogenetic tree. They seem to coevolve consistently with many other genes and are thus fit to represent the evolution of the genome as a whole.

Gene duplications are common across all biological entities, and crAss-like viruses are no exception. While some are recent and found in a small number of genomes, a few are ancient and widespread, suggesting relevance in the evolutionary success of the virus lineage. More specifically, it can be easily speculated that the duplication event involving RepL possibly lends to a reproductive advantage, since such gene is known to increase the number of copies of the plasmid it is found in [43]. Still, in-depth studies are needed to confirm this. Another duplication event found in more than one genome is the one concerning the Reference_crAssphage.1_74 ORF, found in six closely related genomes. This protein has not been characterized and our functional prediction has not been able to unequivocally predict a putative function, though a weak hypothesis has emerged that this could be a structural protein with a trans-membrane region, and potentially kinase-like activity. Transposing elements too are ubiquitous, and one, a HNH endonuclease, has made its way into a crAssphage lineage. It is likely too early to have an idea about its impact on the bacteriophage biology, as this insertion appears to be very recent and specifically localized into a few individuals.

Metaviromics has allowed to investigate more in detail the ecosystem distribution of phages transcending the biases of classical isolation-based studies. The fast technological development of the high-throughput sequencing will allow to overcome the limitations encountered in the study of crAss-like viruses and other metagenomic data: new sequencing techniques can reduce—or eliminate—the need of computationally intensive assembly programs. For example, the application of long read-producing sequencing methods, such as Oxford nanopore [51], would greatly improve the quality of the assembly. While annotation issues are more difficult to address, more investigations, such as 3D structure analysis, are becoming possible. Furthermore, the newly found possibility to grow crAssphage in vitro might open the door to experimental annotation of its proteins [51].

## Figures and Tables

**Figure 1 viruses-12-01035-f001:**
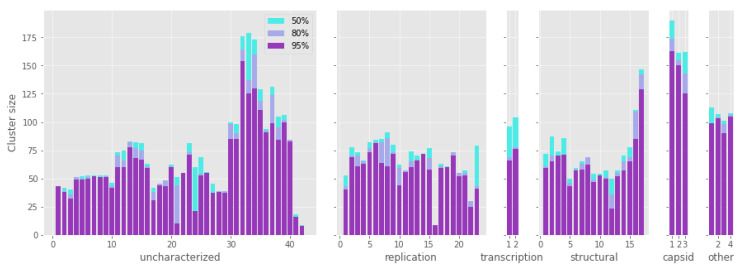
Number of sequences in each group of protein homologs (cluster size) according to functional categories. The three color bars refer to the different similarity thresholds applied to filter alignments (50%, 80%, and 95% respectively). The capsid proteins are the most frequently present in the crAss-like contigs. See Appendix A for the names of all homologous groups of proteins and values for all statistics.

**Figure 2 viruses-12-01035-f002:**
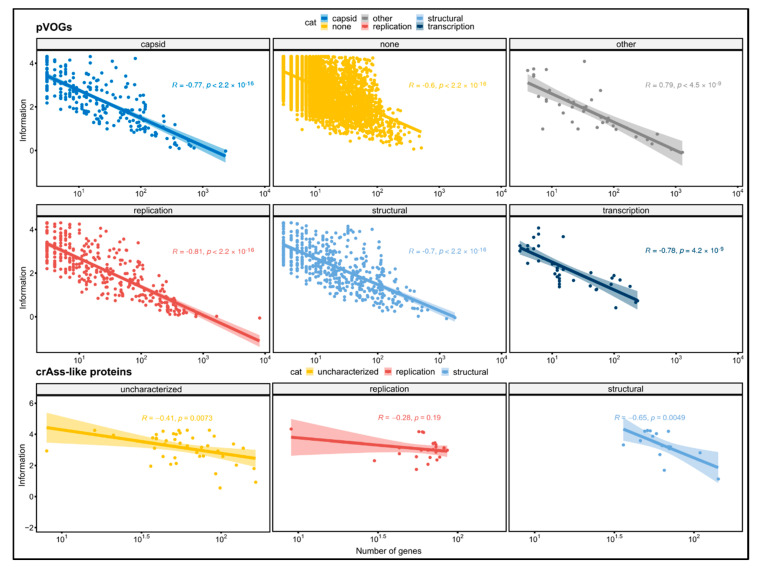
Correlation between the logarithm of the number of sequences in a protein family and the Shannon information content of the positions in the protein sequence alignment. Inverse correlations were obtained both for crAss-like viruses and pVOGs. For proteins in crAss-like viruses, only functional classes having more than four genes are reported.

**Figure 3 viruses-12-01035-f003:**
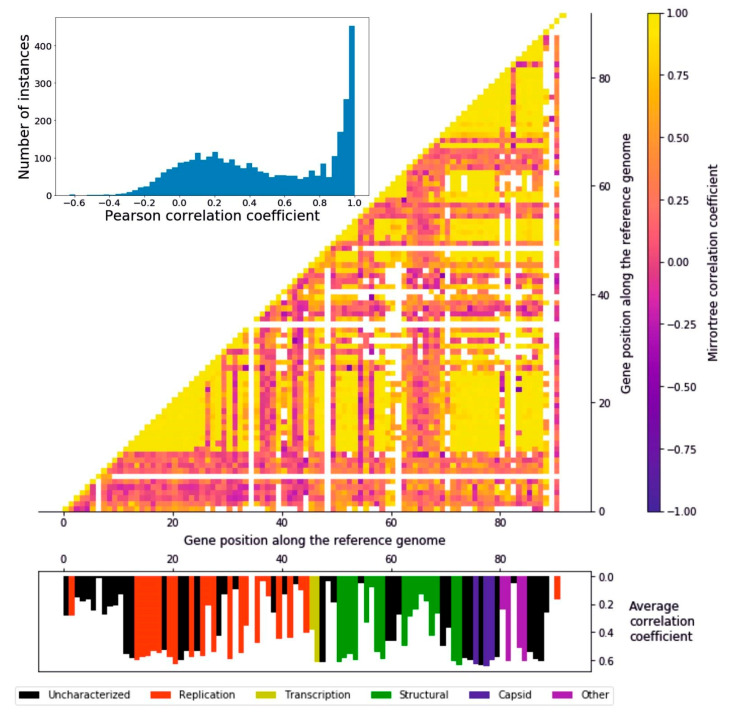
Mirrortree algorithm applied to the homologous groups of proteins. Histogram representing the distribution of the pairwise Pearson’s r coefficients. A great number of genes appear to be coevolving. Heatmap of the Pearson correlation coefficient of each protein with any other. Along the x and y axis the 92 ORFs identified in the reference crAssphage genome are represented. The interactions between clusters sharing less than five sequences were colored in white, in order to avoid confusion due to a low size. Histogram representing the average correlation coefficient of every protein represented in their position on the reference genome. The different colors represent the six functional groups.

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
