# Peer review of "Evolutionary Study of the Crassphage Virus at Gene Level"

_viruses, 2020, doi:10.3390/v12091035_

Round 1

Reviewer 1 Report

Rossi et al. provide an in depth analysis on the genes encoded within crAss-like viruses. After robustly assigning putative proteins into clusters and attempting to disentangle genetic duplication events, they show that crAss-like protein conservation is negatively correlated with cluster size, and expand these findings to other viral protein families by utilizing pVOGs. By pair-wise correlation calculations, they reveal two large blocks of  co-evolving genes. These results provide important information on the evolutionary history of crAss-like viruses, and more importantly suggest that the capsid genes should be used to reconstruct species phylogenies.

I think the paper is very nicely written, and only have a few minor suggestions:

  • In M&M the authors state that a collated set of crAss-like sequences were collected. Contigs derived from PRJNA510571 are well described in the reference. For other accessions (e.g. SRX023429) it is not immediately clear if the authors assembled these contigs themselves or retrieved these from another study. I think it would be nice to have a little more elaboration on the different approaches that were taken to generate the dataset.
  • Line 135, Podoviridae should be in italics.
  • 'x' at line 153
  • Line 228, the second statement is in my opinion not an explanation for the overall negative correlation shown, but rather explanatory for a lower correlation effect in the crAss-like protein families versus the pVOG dataset.
  • Line 283: There is no assigned 'B' panel in figure 3.
  • Supplemental information: There is a number of typo's in the text (e.g. line 53, line 78)
  • Supplemental information: Lines 55 - 57, it would be valuable information to know how much longer ORF 91 was than ORF 45, and subsequently in how many contigs the duplication was verified.
  • The length difference between both ORFs could imply that additional trimming of the alignment would be beneficial and/or required for phylogenetic reconstruction. From M&M it seems that no trimming was performed. Could the authors elaborate here if the alignment needed manual trimming or not ?
  • Supplemental figure S2, it's very difficult to interpret the bootstrap values. It could be beneficial to either have these colored according to a gradient, or only label nodes exceeding 70% or similar.

Author Response

Rossi et al. provide an in depth analysis on the genes encoded within crAss-like viruses. After robustly assigning putative proteins into clusters and attempting to disentangle genetic duplication events, they show that crAss-like protein conservation is negatively correlated with cluster size, and expand these findings to other viral protein families by utilizing pVOGs. By pair-wise correlation calculations, they reveal two large blocks of  co-evolving genes. These results provide important information on the evolutionary history of crAss-like viruses, and more importantly suggest that the capsid genes should be used to reconstruct species phylogenies.

I think the paper is very nicely written, and only have a few minor suggestions:

  • In M&M the authors state that a collated set of crAss-like sequences were collected. Contigs derived from PRJNA510571 are well described in the reference. For other accessions (e.g. SRX023429) it is not immediately clear if the authors assembled these contigs themselves or retrieved these from another study. I think it would be nice to have a little more elaboration on the different approaches that were taken to generate the dataset.

In order to clarify this point, the text has been modified and details have been added in the new version of the manuscript, including reference to the Yutin paper (ref #13) that first presented these sequences (page 3, lines 94-96) as follows:

including a selection of data obtained from the Sequence Read Archive (SRA) as previously described (PRJNA510571) retrieved from previous works [11,13]”

  • Line 135, Podoviridae should be in italics. Done as requested.
  • 'x' at line 153. The letter was removed.
  • Line 228, the second statement is in my opinion not an explanation for the overall negative correlation shown, but rather explanatory for a lower correlation effect in the crAss-like protein families versus the pVOG dataset.
    The reasons explaining the negative correlation between the diffusion and the conservation degree of homologous clusters of proteins were separated as suggested. The text on page 6, lines 238 and following was modified as follows:

There are several possible explanations for this negative correlation. First, widespread sequences may be more divergent than sequences with a narrow occurrence distribution. Secondly, while pVOGs are built from proteins found across all the prokaryotic virus groups, the homologous groups of proteins built in this study only include sequences from a closely related group of contigs, possibly representing a single viral family. Thirdly, the high threshold of 80% query coverage used in the BLASTp has surely raised the degree of conservation between the detected homologous proteins that formed the crAss-like clusters. For these reasons, The most likely explanation for this negative correlation is a sampling effect where many, widespread sequences may contain more diversity than few sequences with a narrow occurrence distribution. The stronger correlation coefficients of pVOGs relative to the homologous groups of proteins in crAss-like viruses that were built in this study is likely due to other factors. Firstly, pVOGs are built from proteins found across all the prokaryotic viruses and include several homologous groups with a wide distribution that strongly contribute to the low correlation coefficient (note the difference in scale of the X-axes in Figure 2). Conversely, the homologous groups of proteins built in this study only include sequences from the relatively closely related group of crAss-like viruses, which have been proposed to represent a single viral family. Thus, no homologous groups are observed with more than a few hundred sequences and the correlation coefficient is less extreme than for the pVOGs. Secondly, the high threshold of 80% query coverage used in the BLASTp has raised the degree of conservation between the detected homologous proteins that formed the crAss-like clusters. For this same reason, the average Shannon information content of crAss-like proteins is rather high; both the mean and the median lie between 3.0 and 3.5 bits, compared to a maximum of about 4.32 bits.

  • Line 283: There is no assigned 'B' panel in figure 3.
    The text was changed:

(Figure 3b)

  • Supplemental information: There is a number of typo's in the text (e.g. line 53, line 78)
    Several typos were corrected:
    In line
    34
    "... other kinds of viruses [2]. After the duplication ..."
    In line 38

    ... expression [4,5]. A frequent destiny...”

In line 42

... size [6,7]. The gene relative ...”
In line 54

a total of three gene duplications hashave been newly detected…”
In line 61

This protein has not been characterized and our functional prediction has not been able to unequivocally predict unequivocally a putative function”

  • Supplemental information: Lines 55 - 57, it would be valuable information to know how much longer ORF 91 was than ORF 45, and subsequently in how many contigs the duplication was verified.
    The sequence lengths in amino acids were added in lines 45-46, and the following sentence was rephrased:

The latter sequence (219 aa) is longer than the former (185 aa); the presence of two paralogs with similar lengths this difference in length between the two paralogs was verified in a total of 16 many other contigs.”

  • The length difference between both ORFs could imply that additional trimming of the alignment would be beneficial and/or required for phylogenetic reconstruction. From M&M it seems that no trimming was performed. Could the authors elaborate here if the alignment needed manual trimming or not ?
    The comment was implemented. On page 3, line 127, the explanation whether the alignment needed to be trimmed was added as follows:

“For every homologous group of proteins a ML tree was built with IQ-TREE v1.6.1 [30] with standard settings and 1000 replicates ultrafast bootstrap [31]. For this analysis, the dUTPase homologous group was manually split into two halves, using AliView v1.19 [32]. Despite the difference in length among the sequences in the RepL homologous group, the alignment did not need manual trimming, as it does not present badly aligned regions which would impair the phylogenetic reconstruction. Furthermore, the phylogenetic tree reconstruction software treats gaps as not containing information, and as such the presence of a well-aligned subset of longer sequences does not have an impact on the overall quality of the reconstruction.

  • Supplemental figure S2, it's very difficult to interpret the bootstrap values. It could be beneficial to either have these colored according to a gradient, or only label nodes exceeding 70% or similar.

The supplementary figure S2 was modified as suggested.

Reviewer 2 Report

Bacteriodes, an obligate anaerobe, probably constitutes most of the bacteria in the primate gut. These bacteria are difficult to study. Thus their phages are not well characterized. The authors have used all of the available DNA sequence data for crAssphage to assemble a 97 kb genome, organized with gene functions as one might expect. This is an impressive piece of work.

line 57: eliminate "nothing but."

line 71: "by" should be "of."

line 313: "it was" should be "was it."

Author Response

Bacteriodes, an obligate anaerobe, probably constitutes most of the bacteria in the primate gut. These bacteria are difficult to study. Thus their phages are not well characterized. The authors have used all of the available DNA sequence data for crAssphage to assemble a 97 kb genome, organized with gene functions as one might expect. This is an impressive piece of work.

line 57: eliminate "nothing but.". Done as requested.

line 71: "by" should be "of.". Done as requested.

line 313: "it was" should be "was it.". Done as requested.
